# Physical drivers of the summer 2019 North Pacific marine heatwave

Dillon J. Amaya [1,2✉], Arthur J. Miller[3], Shang-Ping Xie [3] & Yu Kosaka [4]

Summer 2019 observations show a rapid resurgence of the Blob-like warm sea surface temperature (SST) anomalies that produced devastating marine impacts in the Northeast Pacific during winter 2013/2014. Unlike the original Blob, Blob 2.0 peaked in the summer, a season when little is known about the physical drivers of such events. We show that Blob 2.0 primarily results from a prolonged weakening of the North Pacific High-Pressure System. This reduces surface winds and decreases evaporative cooling and wind-driven upper ocean mixing. Warmer ocean conditions then reduce low-cloud fraction, reinforcing the marine heatwave through a positive low-cloud feedback. Using an atmospheric model forced with observed SSTs, we also find that remote SST forcing from the central equatorial and, surprisingly, the subtropical North Pacific Ocean contribute to the weakened North Pacific High. Our multi-faceted analysis sheds light on the physical drivers governing the intensity and longevity of summertime North Pacific marine heatwaves.

[1] Cooperative Institute for Research in Environmental Sciences, University of Colorado Boulder, Boulder, CO, USA. [2] Department of Atmospheric and Oceanic Sciences, University of Colorado Boulder, Boulder, CO, USA. [3] Scripps Institution of Oceanography, University of California San Diego, San Diego, CA, USA. [4] Research Center for Advanced Science and Technology, The University of Tokyo, Tokyo, Japan. ✉email: dillon.amaya@colorado.edu

I n winter 2013/2014 upper ocean temperatures in the North-east Pacific were remarkably warm over a large area, with peak values near 2.5 °C or three standard deviations above normal. The extraordinary magnitude and persistence of these anomalies posed a significant threat to regional marine ecosystems[1], earning this pattern the moniker The Blob in the scientific literature and media[2]. Over the subsequent months, The Blob anomalies spread along the western North American coastline as the result of shifting atmospheric forcing, generating significant coastal warming and unprecedented marine impacts during the summer of 2014 and into winter 2014/2015[3–5]. As a result, the 2013–2015 North Pacific warm anomalies have since generated significant scientific interest into the origin, persistence, probability, and intensification of so-called marine heatwaves[4–10].

Half a decade later, upper ocean temperatures near the Gulf of Alaska have risen to unprecedented levels (Fig. 1), leading to significant public concern that The Blob and its devastating impacts have once again returned. However, an important distinction between the recent temperature anomalies (hereafter referred to as Blob 2.0) and 2013/2014 temperature anomalies (hereafter referred to as Blob 1.0) is that Blob 2.0 has primarily intensified during the summer season. The Blob 1.0 originated in

the winter and was the result of a resilient atmospheric ridge in the Northeast Pacific, which weakened the climatological Aleutian Low and related surface winds. The reduced wind forcing decreased Ekman advection of cold water from higher latitudes and reduced wind-generated upper ocean mixing, which allowed surface heat fluxes to significantly warm the upper ocean[2].

While stalled atmospheric high-pressure systems have been linked to the development of other wintertime marine heat-waves[10], it is not clear that this is a viable forcing mechanism in the summer when the mean circulation itself is dominated by a ridge—the North Pacific High. For example, an anomalous summertime atmospheric ridge would tend to strengthen the background mean circulation (i.e., the surface westerlies) when superposed on the climatological North Pacific High. This would increase surface evaporation and wind-generated upper ocean mixing. As a result, we might actually expect a series of anomalous summertime ridging events to cool the Northeast Pacific Ocean.

In addition, studies have shown that the remarkable persistence of the winter 2013/2014 Northeast Pacific ridge was partly the result of remote sea surface temperature (SST) forcing through atmospheric teleconnections[4,8,11,12]. There has been considerable research on SST-forced teleconnections to the North Pacific during boreal winter when important climate modes like the El Niño Southern Oscillation (ENSO) tend to peak[13,14], but there has been comparably less work on the influence of remote SST on North Pacific ocean-atmospheric variability during the summer[15,16]. Further, while local air–sea feedbacks were not thought to be an important sustaining mechanism for Blob 1.0, they may be a significant driver of the Blob 2.0 due to the pervasiveness of marine stratocumulus clouds during boreal summer. An analysis of the physical mechanisms behind Blob 2.0 and the extent to which remote and/or local SST forcing may be playing a role would improve our understanding of the development, evolution, and persistence of summertime marine heatwaves in this region.

In this study, we use gridded reanalysis products combined with satellite-derived observations to investigate the physical drivers that led to the rapid intensification of the summer 2019 Northeast Pacific SST anomalies. We find that Blob 2.0 primarily results from an anomalous weakening of the North Pacific High, which significantly reduces wind-driven upper ocean mixing, producing a record shallow mixed layer depth. As a result, strong downward surface heat fluxes mixed over an anomalously thin mixed layer volume, resulting in record warming for this region. We then use a suite of SST-forced atmospheric model simulations to conduct a near real-time attribution analysis of the North Pacific atmospheric circulation anomalies. We show that, while the magnitude of North Pacific High weakening primarily results from internal atmospheric variability, the circulation anomalies are reinforced in time and space by atmospheric teleconnections originating from both the tropics and, surprisingly, the subtropical North Pacific. In addition, we present satellite and model evidence that surface warming associated with Blob 2.0 significantly reduce low-cloud fraction in the North Pacific, suggesting an important role for local air–sea feedbacks in amplifying/sustaining the summer 2019 marine heatwave.

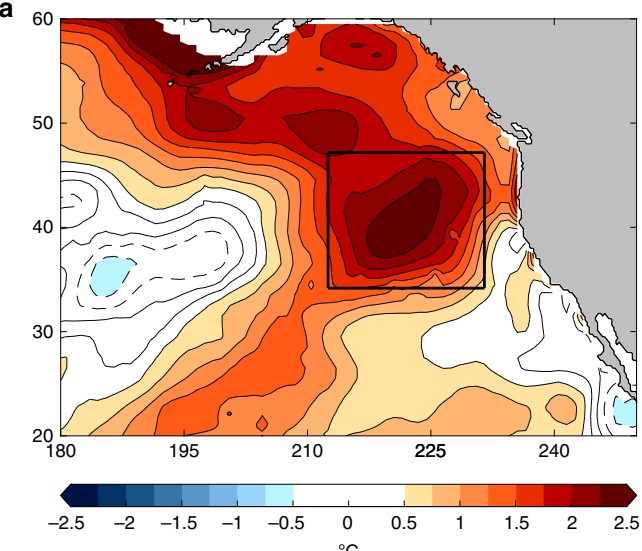

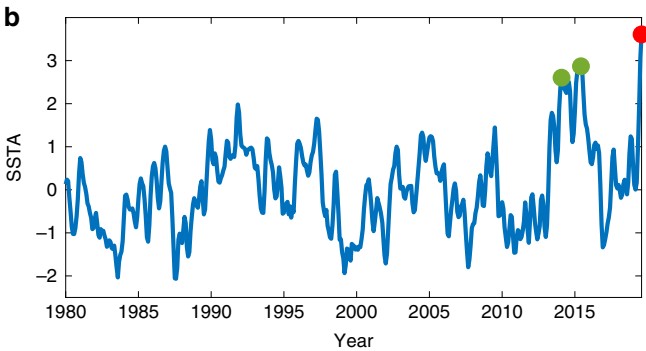

**Fig. 1 The Blob 2.0 in an ocean reanalysis product. a** Five-meter ocean temperature anomalies (°C) averaged for June–August (JJA) 2019 in Global Ocean Data Assimilation System (GODAS). **b** Time series of normalized monthly mean sea surface temperature anomalies area-weighted averaged in black box above, smoothed with a 3-month running mean for the period 1980–2019. Red dot marks JJA 2019. Green dots mark the two peaks of Blob 1.0 averaged in January–March 2014 and May–July 2015, respectively.

## Results

**Drivers of Blob 2.0 in observational analyses.** To investigate the driving mechanisms behind Blob 2.0, we begin by inspecting June–August (JJA) 2019 averaged SSTAs in the North Pacific (Fig. 1a). Similar to 2013/2014, ocean reanalysis data sets show a Blob of anomalous ocean warming off the United States west coast, with a peak magnitude of over 2.5 °C at the center

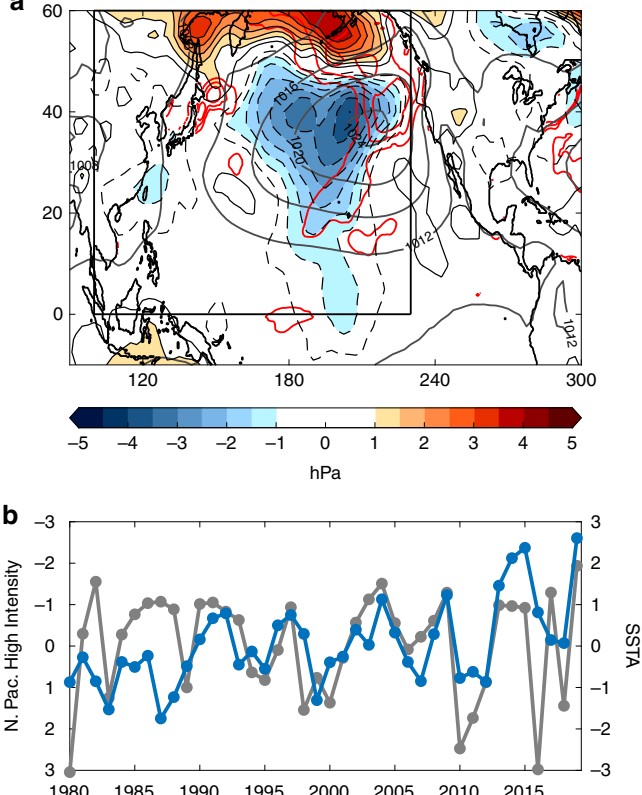

**Fig. 2 Atmospheric circulation anomalies during summer 2019. a** Detrended sea level pressure anomalies (SLPAs; hPa; shading) from atmospheric reanalysis averaged June–August (JJA) 2019 (shading and thin contours). Heavy dark contours represent JJA SLP climatology. Red contours are sea surface temperature anomalies (SSTAs) from Fig. 1a (starting at 1 °C with 0.5 °C interval). Black box represents area of comparison for Best Member results in Fig. 5d–f. **b** Time series of the North Pacific High Intensity index (gray/left y-axis; hPa; negative = weaker; see "Methods") and normalized JJA SSTA area-weighted averaged in the black box shown in Fig. 1a (blue/right y-axis) for the period 1980–2019. Note the inverted left y-axis.

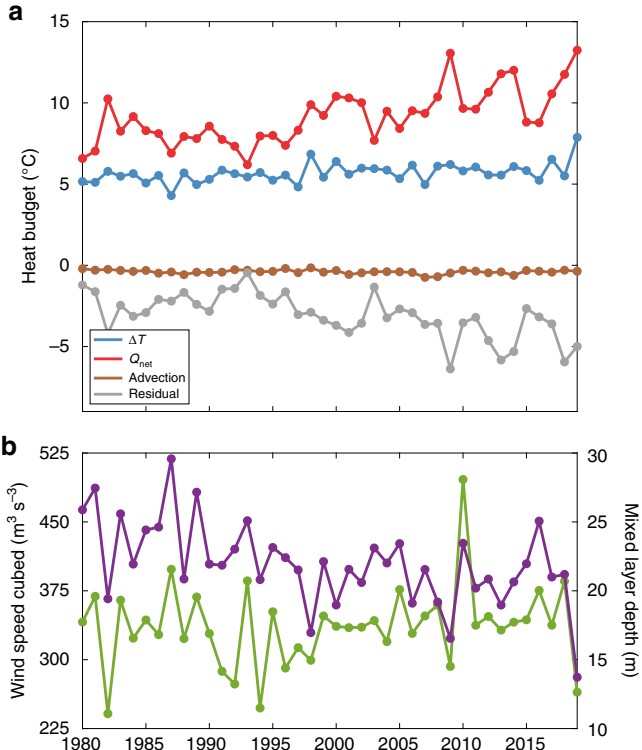

**Fig. 3 Mixed layer heat budget estimated in ocean reanalysis. a** August minus May temperature change (°C; blue) and the respective budget terms contributing to this change. Budget terms include total surface heat flux (red), horizontal advection (brown), and entrainment/residual (gray). **b** June–August (JJA)-averaged reanalysis wind speed cubed (green/left y-axis; $m^3 s^{-3}$) and mixed layer depth (purple/right y-axis; m) area-weighted averaged in black box seen in Fig. 1a. All time series are for the time period 1980–2019.

(~3.5 standard deviations above normal). In order to put these anomalies into historical context, we show an area-weighted average of normalized SSTAs in the domain 34 °N– 47 °N, 147 °W–128 °W from January 1980 to August 2019 (Fig. 1b). Within this area, the 2019 summertime SSTAs (red dot) are not only larger than the 2013–2015 marine heatwave (green dots), but they are also the warmest in at least the last 40 years.

Atmospheric forcing is a key component of SST variability in our region of interest[17]. Therefore, it is essential to assess the state of the North Pacific atmosphere during this time period. To accomplish this, Fig. 2a shows JJA 2019 averaged sea level pressure anomalies (SLPAs). Here, we see a dipole of SLPAs reminiscent of the North Pacific Oscillation (NPO)[18,19], with a lobe of positive anomalies over the Aleutians and broad negative SLPAs extending from ~45 °N to as far south as the equator. Superposed on the mean state (dark gray contours), these anomalies would tend to weaken the summer North Pacific High and the corresponding surface winds. Indeed, when calculating a summertime North Pacific Subtropical High Intensity index (see "Methods"), we find that the surface circulation is the weakest it has been in the last 40 years (Fig. 2b). The significant interannual correlation between the North Pacific High Intensity index and JJA-averaged SSTAs in the Blob 2.0 region ($R = -0.39$; $R = -0.48$

detrended, both 95% significant) suggests that a weakened mean state is a primary driver of summer marine heatwaves in this region.

What impact does this weakened North Pacific High have on the upper ocean? To answer this question, we estimate a mixed layer heat budget for a volume that is bounded horizontally by the black box in Fig. 1a, and vertically from the surface to the bottom of the mixed layer. The upper panel of Fig. 3 shows the May to August temperature change for each year from 1980 to 2019 in the Global Ocean Data Assimilation System (GODAS) ocean reanalysis, along with the budget terms that contribute to this change. We analyze the May–August temperature tendency because Blob 2.0 grew most rapidly over these months (not shown). During the summer, temperature changes in this region primarily represent a balance between warming induced by the net surface heat fluxes (red) and cooling associated with entrainment at the bottom of the mixed layer (e.g., residual; gray). The advective heating term (brown) is negligible, which is not surprising given the size of our mixed layer volume and the relatively weak eddy kinetic energy found in this part of the North Pacific[20]. Based on this analysis, the May to August temperature tendency (blue) shows a record warming of 7.9 °C for 2019. This difference anomaly (2.2 °C) is about 40% more than the mean and is almost entirely due to record positive SSTAs in August (Supplementary Fig. 1).

The remarkable warming is primarily the result of strongly positive net surface heat fluxes (Fig. 3a red), which are dominated by net shortwave radiation fluxes with a smaller but important

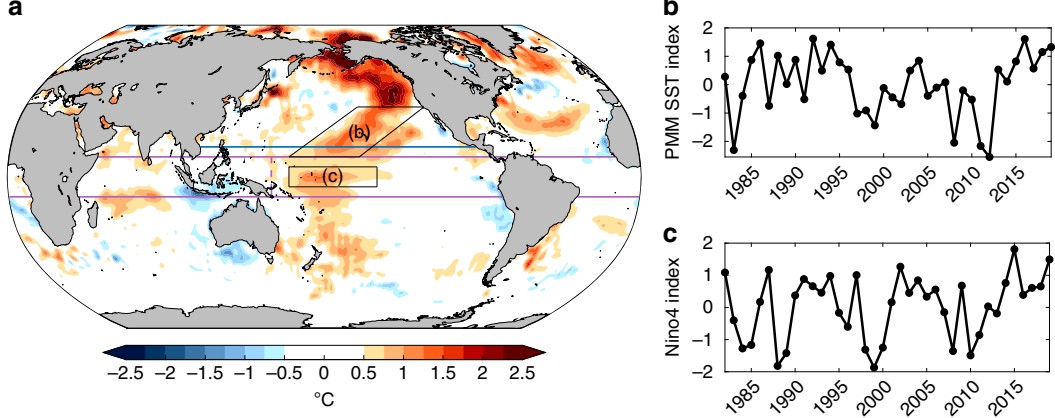

**Fig. 4 Summer 2019 surface temperature anomalies and model domains. a** Global observed sea surface temperature anomalies (SSTAs; °C; shading) used to force our atmospheric model experiments. Domain for tropical SST-forced experiment is 10 °S–10 °N at all longitudes (i.e., equatorward of solid purple lines). Domain for North Pacific SST-forced experiment is poleward of 15 °N to the coastlines (i.e., poleward of solid blue line). Vertical dashed purple line represents separation point for tropical Indian and tropical Pacific Ocean experiments (Supplementary Fig. 4). Black boxes represent areas of interest in (**b**) and (**c**). **b** Normalized June–August (JJA)-averaged Pacific Meridional Mode SST index (see "Methods"). **c** Normalized JJA-averaged SSTA area-weighted averaged in the Nino4 region. Each time series is shown for the period 1980–2019.

contribution from the latent heat flux (see Flux terms in Supplementary Fig. 2). An increase in net shortwave radiation suggests changes in cloud fraction may be important. We will explore this connection in more detail in the following sections. A reduction in latent heat flux is consistent with a weakened North Pacific High. The weakened High would tend to reduce surface wind speeds and wind-driven evaporation, thereby warming the upper ocean. This is consistent with the location of the warmest Blob 2.0 SSTAs relative to the anomalous surface winds implied by the negative SLPAs (red contours; Fig. 2).

The surface winds are further related to the ocean mixed layer depth through the wind speed cubed (Fig. 3b green), which is a measure of the power imparted by the atmosphere to the ocean for turbulent mixing[21]. In this case, the JJA-averaged wind speed cubed in our box shows a significant reduction in wind-driven upper ocean mixing during this time period. As a result, the mixed layer depth (purple) was 62% shallower than average, a record over our analysis time period. Inspection of monthly mean mixed layer depths prior to summer 2019 reveals that the mixed layer first shoaled significantly in April, which would also precondition the upper ocean in this region to enhanced warming during JJA.

Given that the summertime climatological mixed layer depth in the Blob 2.0 region is very thin relative to other seasons, it is important to assess the direct influence of these mixed layer depth perturbations on the SST tendency itself[22]. Isolating the mixed layer perturbation effect shows that it dominates the interannual variability of the JJA-averaged surface heat flux terms in the Blob 2.0 region (with the exception of the sensible heat flux, which is small; see MLD terms in Supplementary Fig. 2). However, for the JJA 2019 event, the combination of positive net surface heat fluxes and an extremely thin mixed layer volume work in concert to explain the rapid intensification of Blob 2.0 during summer 2019.

**Remote SST forcing.** Building on the understanding that the weakened North Pacific High was a significant contributor to the evolution of Blob 2.0, it is interesting to investigate the forcing mechanisms that led to the development of these atmospheric circulation anomalies. Inspection of monthly mean SLPA maps from atmospheric reanalysis data reveals that the negative SLPAs seen in Fig. 2 were not a transient pattern, but a persistent feature of the North Pacific atmosphere from April to August 2019 (not shown). Such multi-month consistency suggests that remote or

local SST forcing may play a vital role in shaping and reinforcing these anomalies. This is interesting given remote SST forcing of the North Pacific ocean-atmosphere system tends to be most coherent in boreal winter, when ENSO-related SSTAs are strongest and the stronger westerly winds in the midlatitudes provide a stronger Rossby wave source and wave guide[23,24].

However, there is some observational evidence that central Pacific (CP) ENSO events trigger an atmospheric teleconnection to the North Pacific during boreal summer[15]. In particular, studies suggest that CP ENSO events may be linked to NPO variability on decadal timescales[25,26]. The NPO spatial pattern bears a striking resemblance to the JJA 2019 SLPAs seen in Fig. 2a, which is further supported by the strong correlation between our North Pacific High Intensity index and the summertime NPO index ($R = 0.65$; see "Methods"). In addition, recent work has focused on the influence of subtropical SSTAs associated with the Pacific Meridional Mode (PMM)[27,28] on the position of the mean intertropical convergence zone (ITCZ) in boreal summer. A PMM-driven shift of the ITCZ has been shown to produce a large-scale atmospheric circulation response that spans the breadth of the subtropics and projects on the North Pacific High[29,30]. This process, termed the summer deep convection (SDC) response, then projects onto mid-latitude SSTs through changes in surface heat fluxes.

Both the PMM SST index[27] (see "Methods") and CP ENSO (as estimated by the Nino4 region) show above average values during summer 2019 (Fig. 4). Therefore, it is possible that these SST-forced effects may significantly contribute to the weakened North Pacific atmospheric circulation. We test this hypothesis by producing a comprehensive suite of SST-forced atmospheric general circulation model (AGCM) ensembles. Specifically, we conduct three sets of AGCM experiments, each integrated with observed SSTs from January 2018 to August 2019 and consisting of 20 ensemble members. The three experiments are global SST forcing, tropical (10 °S–10 °N) SST only forcing, and North Pacific (>15 °N) SST only forcing (see "Methods" for more details). The SSTA patterns used to force the various experiments during JJA 2019 can be seen in Fig. 4.

Figure 5a–c shows the ensemble mean SLPAs results for each AGCM experiment during JJA 2019. In the global SST-forced simulations (Fig. 5a), the ensemble mean produces a cyclonic SLPA pattern that broadly resembles the weakening of the North Pacific High seen in Fig. 2. The weakened High is connected to a

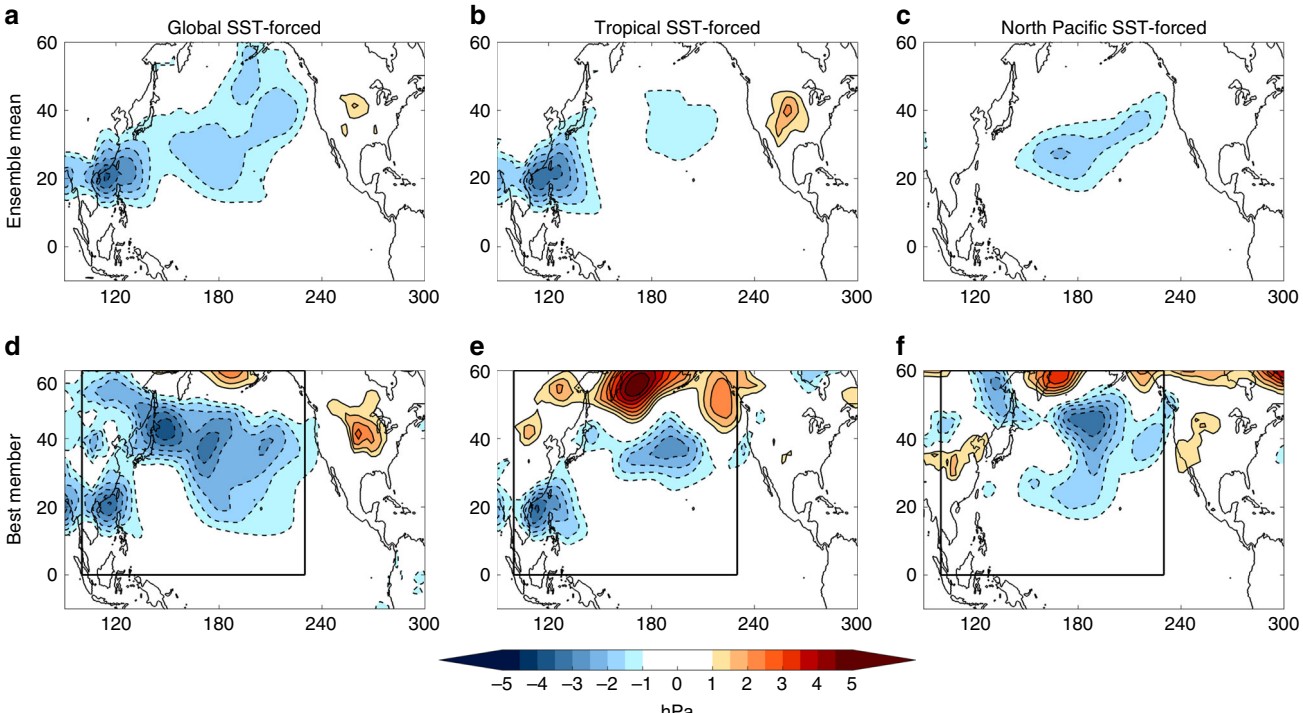

**Fig. 5 Simulated sea level pressure anomalies for summer 2019. a–c** Ensemble mean June–August (JJA) 2019 sea level pressure anomalies (SLPAs; hPa; shading) for the Global, Tropical, and North Pacific sea surface temperature forced experiments, respectively. **d–f** Single Best Member match for each ensemble of simulations based on pattern correlation of model and atmospheric reanalysis SLPAs in black box (see Fig. 2). Simulated anomalies shown are significant at 95% for a Student's *t* test.

second, more pronounced center of negative SLPAs north of the Philippines near Taiwan. This feature is also seen in atmospheric reanalysis data (although much weaker) and is associated with large precipitation anomalies that suggest enhanced deep convection in this region (Supplementary Fig. 3). In contrast, the northern, anticyclonic SLPA lobe seen in Fig. 2 is not reproduced by the model, indicating that these anomalies are primarily the result of internal atmospheric variability.

The global SST-forced SLPA results are further decomposed into the tropically- and North Pacific-forced components (Fig. 5b, c, respectively). Here we see that both tropical and North Pacific SST contribute to the weakening of the North Pacific High. In particular, the tropical SST-forced ensemble mean results are consistent with the observed SLPA spatial structure, if not in magnitude. In order to isolate which tropical ocean basin is responsible for this atmospheric response, we generate a separate set of 20-member ensembles for the tropical Pacific Ocean-only and the tropical Indian Ocean-only. We find that the weakened North Pacific High in this AGCM experiment is almost entirely due to the aforementioned CP El Niño conditions (Supplementary Fig. 4, tropical Pacific), with minimal contribution from Indian Ocean SST forcing. According to the monthly Nino4 index, these CP El Niño conditions were first established in winter 2018/2019 and then persisted into our boreal summer 2019 analysis period (not shown).

Similar to the tropical Pacific runs, the North Pacific SST-forced simulations produce a cyclonic circulation centered around 30 °N that resembles the atmospheric response to a PMM-driven northward shift of the ITCZ (see Fig. 3 and ref. [28,30]). The northward shifted ITCZ is evident in the AGCM precipitation data and is qualitatively consistent with observations (Supplementary Fig. 3). This suggests that the 2019 PMM event may have produced an active SDC response during this time period, which would have the potential to reinforce the

subtropical SSTAs through wind-evaporation-SST feedback[30,31]. However, we note that our North Pacific-only AGCM simulations cannot explicitly rule out the influence of local SST forcing on the SLPA structure seen in Fig. 5c. These mechanisms, which are separate from the SDC response described above, will be explored further in the following section.

While the AGCM ensemble mean results indicate that atmospheric teleconnections associated with CP El Niño and the PMM contributed to the persistence of the weakened North Pacific High, they cannot fully explain the magnitude of the anomalies in the atmospheric reanalysis. For example, the observed North Pacific High Intensity index value for JJA 2019 is −1.94 hPa (Fig. 2b), while the AGCM ensemble mean simulations produce values of −1.58 hPa, −1.01 hPa, and −1.07 hPa for the global, tropical, and North Pacific SST runs, respectively. On one hand, we may not expect the model to fully reproduce the observations given that the real world is a single realization and the model ensemble mean merely represents the potential strength of the SST-forced component without the benefit of internal noise to enhance (or obscure) the forced signal. However, it is important to analyze the ensemble spread to investigate whether the forced SST signal plus some internal atmospheric state more realistically reproduces the atmospheric reanalysis.

Figure 5d–f shows the best simulation of the 20 members from each experiment that most closely matches Fig. 2a based on a pattern correlation in the black box. Each AGCM ensemble produces a member that more closely captures the spatial structure and magnitude of the atmospheric reanalysis, with pattern correlations of $R = 0.50$, $R = 0.57$, and $R = 0.71$ for the global, tropical, and North Pacific SST runs, respectively. Note that these same three individual ensemble members are also selected when comparing a region that does not include the Southwest Pacific. In addition, the global, tropical, and North

Pacific SST experiments have a total of nine, three, and three ensemble members, respectively, that produce JJA 2019 North Pacific High Intensity index values that are equal to or more negative than the JJA 2019 observed value (−1.94 hPa; Fig. 2b). The fact that individual ensemble members can more accurately capture the observed magnitude of the North Pacific High weakening suggests that internal atmospheric variability is essential to explaining the real-world anomalies; however, it is clear from the ensemble mean results that remote SST forcing also informs their spatial structure and persistence.

**Local SST forcing and air–sea feedbacks.** A final point of inquiry concerns the influence of local SST and air–sea feedbacks in maintaining the 2019 North Pacific marine heatwave. For example, low-cloud feedback is an important amplifying factor for SST perturbations in the subtropical North Pacific, particularly during summertime when marine stratocumulus clouds are most pervasive[32,33]. We investigate this potential feedback by showing low-cloud fraction anomalies for JJA 2019 in both satellite-derived observations and the North Pacific-only AGCM ensemble mean (Fig. 6). Here, satellite observations show a pattern of reduced low-cloud fraction that closely follows the warmest North Pacific SSTAs from the subtropics to the Northeast Pacific. This reduction in low-clouds is associated with an 11.37 W m$^{-2}$ increase in net shortwave radiation in the Blob 2.0 region, which is over twice as large as the surface wind speed-induced latent heat flux (5.09 W m$^{-2}$) during the same time period (Supplementary Fig. 5).

The co-location of negative low-cloud fraction anomalies with positive SSTAs points towards a possible positive feedback. This is confirmed when comparing to the ensemble mean of the North Pacific-only AGCM experiment, which shows a pattern of negative low-cloud fraction anomalies that closely mirrors the observations. These results suggest the 2019 North Pacific marine heatwave was amplified not only by remote SST forcing, but local air–sea interactions as well. Further, a month-by-month comparison of the low-cloud induced change in net shortwave radiation with the latent heat flux suggests that this low-cloud SST feedback was primarily important in maintaining the Blob 2.0 SSTAs in July and August once they had already grown by more than 1 °C (not shown). Whereas, the peak latent heat flux anomalies precede the largest shortwave anomalies by several months, and may be thought of as a triggering mechanism for this event

brought about by the weakened North Pacific High. These results echo an earlier study by Schmeisser et al.[34], which revealed that the mechanisms for the development of these mid-latitude marine heatwaves can differ from those for their maintenance. We encourage future studies on this topic once more data becomes available during the lifespan of Blob 2.0.

Finally, anomalous diabatic heating from the reduced low-cloud cover would tend to further weaken the North Pacific High through the hydrostatic effect on atmospheric pressure[35], which could then couple back to the SST through the surface wind field. Adopting the Lindzen and Nigam[35] framework for the North Pacific equatorward of 40 °N, we find that ~1 hPa of the negative SLPAs shown in Fig. 2a may be attributed to a warmer atmospheric boundary layer driven by positive SSTAs around 20 °N (not shown). This provides a separate opportunity for local SSTs to influence the evolution of the marine heatwave.

## Discussion

Recent observational analyses show a rapid resurgence of the Blob-like anomalies that produced devastating marine ecological impacts in the Northeast Pacific during the winter of 2013/2014[36]. Unlike the original Blob, Blob 2.0 occurred in the summer, a season when relatively little is known about the physical drivers of such events. Our multi-faceted analysis sheds light on these mechanisms with a focus on the origin, intensity, and longevity of summertime North Pacific marine heatwaves. In particular, our results show that the 2019 Blob 2.0 primarily resulted from a weakened North Pacific High, which reduced the strength of the surface winds, resulting in reduced evaporative cooling and wind-driven upper ocean mixing in the Northeast Pacific. Consequently, strong downward surface heat fluxes were mixed over a record minimum mixed layer depth, producing surface warming in excess of 2.5 °C above normal.

Our atmospheric model analysis reveals that the magnitude of the North Pacific High weakening was consistent with internal atmospheric variability; however, the multi-month persistence of these anomalies was aided by remote SST forcing from the central equatorial Pacific and the subtropical North Pacific. Further, our results indicate that a weakened North Pacific High is a primary driver of summertime Northeast Pacific SST extremes on inter-annual timescales. In addition, satellite observations show a reduction in North Pacific low-cloud fraction that is consistent with a positive low-cloud feedback. This is confirmed by our

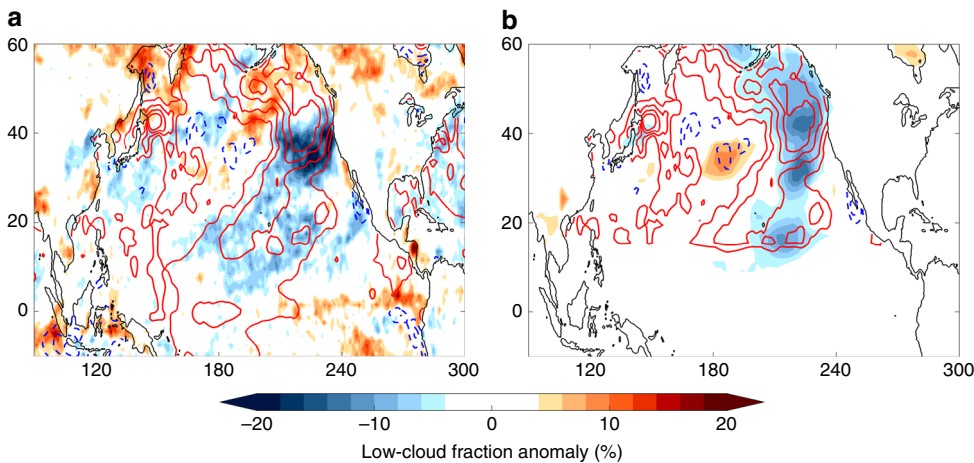

**Fig. 6 Low-cloud fraction anomalies for June–August 2019.** Low-cloud fraction anomalies (shading; %) from **a** satellite-derived observations and **b** ensemble mean of North Pacific sea surface temperature forced experiment. Colored contours represent observed sea surface temperature anomalies (SSTAs; °C) seen by both (**a**) and (**b**), respectively. SSTA contour interval is 0.5 °C with a maximum value of 2 °C. Simulated anomalies shown are significant at 95% for a Student's *t* test.

North Pacific SST-forced model simulation, which suggests that this local air–sea feedback was also important for the maintenance and persistence of the 2019 marine heatwave.

These results highlight the seasonal differences in the relevant mechanisms that lead to the growth and persistence of North Pacific SST extremes and ultimately to their potential impacts. For example, the wintertime Blob 1.0 contended with a climatologically deep ocean mixed layer. This meant surface heat fluxes were mixed over a larger volume, dampening their influence on the overall mixed layer temperature. On the other hand, a deep ocean mixed layer would allow surface anomalies to become trapped below the mixed layer as it shoals in the summer and reemerge at the surface in the following winter when the mixed layer deepens once more[37]. Therefore, Blob 1.0's prolonged impact on marine ecosystems into winter 2014/2015 may have resulted, in part, from this reemergence mechanism. In contrast, the Blob 2.0 SSTAs intensified during the warm summer season, which could lead to short-lived but potentially severe ecosystem impacts if the thermal tolerances of sensitive marine species are exceeded. However, Northeast Pacific mixed layer depths were extremely shallow during this time period (Fig. 3b), so Blob 2.0 persistence into the winter and spring 2020 seasons when coastal biological impacts are most likely to be felt[1] depends on the consistency of atmospheric forcing and strength of local air–sea interactions like low-cloud feedback. Without sustained forcing, it is unlikely Blob 2.0 will produce the same prolonged damage to marine ecosystems that was generated by its predecessor.

We also note the striking multidecadal features evident in many of our time series. In particular, there is a significant positive trend in JJA-averaged Blob 2.0 SSTAs and surface heat flux, as well as a negative trend in mixed layer depths averaged in this region (Figs. 1b and 3). These trends may be related to anthropogenic global warming, which would tend to increase stratification in the upper ocean and dampen ocean mixing[38]. In addition, there also appears to be a decadal transition in the interannual variance of JJA-averaged SSTAs in this region (Fig. 1b). From 1980 to 2000, the detrended summertime interannual variance was 0.24 °C$^2$. From 2000 to 2019, the variance was 0.49 °C$^2$, an increase of 106%. Marine heatwave frequency has been shown to increase due to warming in background SST associated with anthropogenic forcing[7], which could account for these changes. However, the North Pacific climate system experiences robust decadal modulations related to the Pacific Decadal Oscillation (PDO)[39], and the year 2000 roughly coincides with the transition from a positive to a negative phase of the PDO[40]. Therefore, the influence of internal climate variations cannot be completely ruled out. More research is needed on the relative contributions of internal and forced variations to summertime decadal variability in the Northeast Pacific.

Also of interest are the significant SSTAs found north of the Blob 2.0 region in the Gulf of Alaska and in the waters near the Bering Strait (Fig. 1). While these features are not the focus of this study, it is important to realize that they predate the emergence of the Blob 2.0 anomalies discussed here and were accompanied by significant surface air temperature anomalies over Alaska (not shown). This difference in timing suggests the mechanisms that led to these anomalies may differ from those that drove Blob 2.0 and may include factors unique to higher latitudes such as sea ice loss. In addition, while low-cloud SST feedbacks were shown to be important to the intensification of Blob 2.0, the disagreement between satellite observations and our AGCM simulations in the Gulf of Alaska indicates that this process did not significantly contribute to the persistence of these SSTAs into JJA 2019 (Fig. 6). Instead, the easterly surface wind anomalies around 45 °N implied by the SLPAs in Fig. 2a may have bolstered the Gulf of Alaska SSTAs through Ekman advection of warm waters

from lower latitudes. Overall, the 2019 North Pacific marine heatwave offers an opportunity to appreciate the often latitudinally dependent mechanisms that shape large-scale SSTA patterns in this region.

Based on our results and the results of other studies, it will be increasingly important to distinguish between the physical drivers of marine heatwaves that occur during different seasons. Making such distinctions would provide better insight into the potential intensity and longevity of a given event, which would benefit those in fisheries and wildlife management who depend on accurate seasonal forecasts of marine heatwaves to make decisions.

## Methods

**Reanalysis data.** Ocean data used in this study is based on reanalysis output from the National Centers for Environmental Prediction (NCEP) GODAS[41] for the period 1980–2019. This ocean reanalysis product is based on the Geophysical Fluid Dynamics Laboratory (GFDL) Modular Ocean Model version 3 (MOMv.3) with data assimilation of profile information from expendable bathythermographs, moored buoys, and Argo profiling floats. GODAS outputs monthly mean data on a 0.33 latitude × 1 longitude grid with 40 vertical levels. Surface fluxes are provided to MOMv.3 from the NCEP Reanalysis 2. We further utilized NCEP Reanalysis 2 for monthly mean sea level pressure and latent heat flux on ~2 × 2 grid. Monthly mean wind speed data on a 2.5 × 2.5 grid was used from the NCEP Reanalysis 1. Monthly mean SST data is taken from the Optimum Interpolation SST version 2 (OISSTv2), which is provided by the National Oceanic and Atmospheric Administration (NOAA) from September 1981 to present on a 1 × 1 grid. All anomalies reported in this paper are based on monthly mean data after removing long-term monthly averages. Anomalies from the above products are based on a long-term climatology for the period 1982–2018.

**Satellite-derived data.** Low-cloud fraction was calculated from Level 3 satellite-derived monthly means from the Moderate Resolution Imaging Spectroradiometer (MODIS)[42], which is currently available on a 1 × 1 grid from February 2000 to August 2019. MODIS anomalies are based on a long-term climatology for the period 2001–2018.

**Mixed layer heat budget.** The mixed layer heat budget was estimated for a volume bounded by the domain 34 °N–47 °N, 147 °W–128 °W horizontally (black box, Fig. 1a), and the surface to the mixed layer depth vertically. The mixed layer depth is offered as a GODAS model output, and is estimated at each grid-point and each time step as the depth where the buoyancy difference with respect to the surface level is equal to 0.03 cm s$^{-2}$ [41]. The budget was calculated using equation (2) in Cronin et al.[43], with the vertical entrainment terms calculated as a residual. The tendency term represents the mixed layer average temperature difference of August minus May for each year. The budget terms were calculated for the May–August average of monthly mean data. Each term was then been multiplied by 3 months to convert to units of degrees Celsius. GODAS data were used for all components of the calculation except for the decomposition of the net surface heat flux into its contributing terms (Supplementary Fig. 2). See the Potential data errors and uncertainties subsection for details regarding potential errors in our mixed layer heat budget.

**North Pacific High Intensity index.** The North Pacific High Intensity index shown in Fig. 1b is based on the North Pacific Subtropical High (NPSH) strength index developed by Schmidt et al.[44], which measures the average monthly mean NCEP Reanalysis 2 SLP in a 10 × 10 box centered on the NPSH SLP centroid. See ref. [44] for more details. We then show anomalies of this index (hPa) relative to a 1982–2018 climatology in Fig. 1b. We interpret negative (positive) values of the North Pacific High Intensity index as a weakening (strengthening) of the summertime NPSH.

**Pacific Meridional Mode (PMM) SST index.** The PMM SST index is from Chiang and Vimont[27] and is available as monthly means from 1948 to present. This index represents the first SST expansion coefficient of a maximum covariance analysis of SST and surface winds in the Pacific Ocean. See ref. [27] for more details.

**North Pacific Oscillation (NPO) index.** The NPO[18,19] index is calculated as the second EOF of NCEP Reanalysis 2 monthly mean SLPAs in the domain 20 °N–60 °N, 120 °E–80 °W. The JJA-averaged second Principle Component was then used for comparison with our North Pacific High index in the "Results" section.

**Atmospheric General Circulation Model (AGCM) experiments.** We performed experiments using the GFDL Atmospheric Model version 2.1 (GFDL-AM2.1)[45], which is available on a ~2 × 2.5 grid with 24 vertical levels. Using GFDL-AM2.1 we

generate three primary sets of experiments in which we force the lower boundary of the model with the trajectory of OISSTv2 SSTs from January 2018 to August 2019 in different regions. The first 17 months are discarded as spin-up, and June–August 2019 are used for analysis. The three experiments include global SST forcing, tropical (10 °S–10 °N) SST forcing (i.e., equatorward of the solid purple lines, Fig. 4), and North Pacific (>15 °N) SST forcing (i.e., poleward of the solid blue line to the coastlines, Fig. 4).

Outside of the respective forcing regions (e.g., tropics or North Pacific), the atmosphere is forced with a repeating SST climatological seasonal cycle with no anomalies. Each simulation consists of 20 ensemble members, each initialized with slightly different initial conditions and forced with greenhouse gases set to 1860 levels. Anomalies in GFDL-AM2.1 are relative to a 30-year control simulation in which the model is forced at all grid points with a repeating SST climatological seasonal cycle with no anomalies and greenhouse gases set to 1860 levels. For the control, the climatological seasonal cycle is based on the OISSTv2 long-term monthly means for the period 1982–2018.

**Potential data errors and uncertainties.** It is important to note the potential errors and uncertainties that may arise in our analysis based on our chosen data sets and methods. In particular, the heat budget presented in Fig. 3 may suffer from uncertainty as a result of using GODAS ocean reanalysis, which includes data assimilation of in situ ocean measurements, but may also include nonphysical heat sources and sinks to match the model to observations. While GODAS has been successfully implemented in the past when considering large-scale climate variability[2,46], calculating the heat budget offline across long-term average fields may cloud interpretation of the results (particularly the mechanisms that drive variability in the residual) and/or introduce errors into the calculations. Despite these concerns, the spatial consistency of atmospheric circulation and low-cloud fraction anomalies (Figs. 2 and 6) relative to the maximum Blob 2.0 anomalies gives us confidence in our interpretation of this particular event.

Further, a recent study by Fiedler et al.[47] showed that the data used to force our AGCM experiments (OISSTv2) exhibits significant biases relative to in situ measurements and can differ by as much as ±0.1–0.5 °C depending on the region. While these errors are admittedly significant on smaller spatiotemporal scales, the focus of this work is on large-scale ocean-atmosphere variability and its impact on SSTA extremes in excess of 2.5 °C. Therefore, we do not expect these biases to unduly influence the results of this paper, particularly when also considering the potential uncertainties that arise from imperfect model physics in our AGCM. Nevertheless, the known biases in OISSTv2 and our chosen AGCM should be taken into consideration when interpreting our results. We defer to future research to assess the sensitivity of this work to the choice of AGCM and/or observational analysis products.

## Data availability

All data used in this study are available online or from the corresponding author on request. GODAS ocean reanalysis data is publicly available at: https://www.esrl.noaa.gov/psd/data/gridded/data.godas.html. NCEP Atmospheric Reanalysis 1 data is publicly available at: https://www.esrl.noaa.gov/psd/data/gridded/data.ncep.reanalysis.html. NCEP Atmospheric Reanalysis 2 data is publicly available at: https://www.esrl.noaa.gov/psd/data/gridded/data.ncep.reanalysis2.html. OISSTv2 sea surface temperature data is publicly available at: https://www.esrl.noaa.gov/psd/data/gridded/data.noaa.oisst.v2.html. MODIS Level 3 satellite data is publicly available at: https://ladsweb.modaps.eosdis.nasa.gov/. CERES-EBAF surface radiation data is publicly available at: https://ceres.larc.nasa.gov/order_data.php. GPCP precipitation data is publicly available at: https://www.esrl.noaa.gov/psd/data/gridded/data.gpcp.html. The PMM index is publicly available at: https://www.esrl.noaa.gov/psd/data/timeseries/monthly/PMM/.

## Code availability

The data in this study were analyzed with publicly available tool packages in MATLAB. All figures were produced by the authors, also with MATLAB. Scripts are available upon requests.

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

## Acknowledgements

D.J.A is supported by the National Science Foundation Graduate Research Fellowship (NSF; DGE-1144086). A.J.M is supported by the NSF (Coastal SEES OCE1600283 and CCE-LTER OCE1637632). The National Oceanic and Atmospheric Administration (NOAAMAPP; NA17OAR4310106) provided additional funding for this research. S.P.X. is supported by the NSF (1637450). Y.K. is supported by the Japan Society for the Promotion of Science (18H01278 and 19H05703), the Japan Science and Technology Agency through Belmont Forum CRA InterDec, and the Japan Ministry of Education, Culture, Sports, Science and Technology through the Integrated Research Program for Advancing Climate Models. We would like to thank Nate Mantua, Pascal Polonik, Reuben Demirdjian, and Amato Evan for their helpful comments during the course of our study.

## Author contributions

D.J.A. and A.J.M. designed the study with input from S.P.X. and Y.K. D.J.A. performed the analyses and wrote the paper with contributions from all authors.

## Competing interests

The authors declare no competing interests.
