## [Peer Review File · Nature Communications]

Reviewers' comments:

Reviewer #1 (Remarks to the Author):

Physical drivers of the summer 2019 North Pacific marine heatwave—the Blob 2.0 by Amaya et al.

This paper looks at the characteristics and drivers of a recent marine heatwave in the north Pacific. The authors use a heat budget to identify the local processes that lead to the event and bespoke model experiments to identify the influence of remote drivers. The manuscript is well written and relates to a scientifically interesting problem. Before publication however, I feel that a few issues need to be clarified.

A couple of general issues:

- I think there are two issues with the heat budget that at the very least require some discussion. If I understand correctly the budget is conducted over 3 month timesteps. Heat budgets conducted offline and particularly using long term averaged fields are likely to contain substantial errors as the alias high-frequency processes i.e. it is very difficult to close the budget. As such its hard to say what component of the residual related to vertical entrainment. Related to this, I believe the GODAS reanalysis includes data assimilation. This implies non-physical heat sources/sinks to match the model with observations. If these terms are large it would make any interpretation of a heat budget problematic.
- Its important to include an indication of what anomalies are statistically significant in the composite maps for the model experiments.

Specific comments:

65: Im surprised you don't cite Di Lorenzo and Mantua 2016 here

98: are not only more significant ...

What do you mean by significant. To me 'significant' would relate to its impacts, however I presume you are just saying that SSTA was larger.

Fig. 2: it would be very helpful to include a contour showing the core of the MHW (based on Fig 1)

108: Subtropical High Intensity index

The use of the Subtropical High Intensity index seems somewhat circular to me given that the index maximises the combined SLPA and SSTA variance. It would be much cleaner to use an SLPA or wind speed index.

120: the residual would also include a possibly substantial error related to performing the calculations offline using averaged data and from the fluxes associated with the data assimilation.

147 BOREAL winter

Fig 5: Are these also JJA averages, as in Fig 2?

173: that resembles the observed weakening ...

The simulated response is far weaker than observed. Indicating that SST forcing can only account for a small fraction of observed circulation response

174: also seen in observations

In contrast to the above the simulated low over the Philippines is far weaker than in observations-. Size matters – so these discrepancies should be noted

193-195. This may be the case, but your experiments cant rule out that the SLP response is purely

related to local SST forcing, rather than to a teleconnection associated with a shift in the ITCZ

279: Marine heatwaves are projected to become more frequent in a warmer world, which could account for these observed changes.

This sentence doesn't seem to make logical sense. How can future projections account for past changes?

Reviewer #2 (Remarks to the Author):

The manuscript focuses on a recent marine heatwave event in the North Pacific, which occurred in summer 2019, the "BLOB2.0", as opposed to the winter-time event "BLOB1.0" in 2013-2014. The authors conclude that the persistent SST anomalies were driven by a weakening of the North Pacific High pressure system with a significant influence of local SST feedback (through reduced low-cloud coverage). Remote SST variability from both the tropical and subtropical North Pacific is also shown to have contributed to the anomalous weakening of the pressure system. The work is original and significant, with clear motivation and discussion. The data and methodology seem appropriate, although the agreement between Sea Level Pressure anomaly observations and modelling experiments is overstated and not always convincing. This is the main issue of the manuscript, and probably more a question of wording than un-conclusive results. I recommend publication after major revision and my comments are included below.

GENERAL:

- The distinction between observations and model outputs is important. NCEP GODAS is not an observational product. The first figure should include satellite SST anomalies, and not only from OISSTv2 product which has been shown to be significantly biased in recent years (Fiedler et al., 2019).
- Adding all the components of the heat flux budget in supplementary materials would improve the manuscript.
- As stated above, the comparison of observed and modelled SLPAs is not convincing: a few statements are hard to follow when looking at the figures (see below).
- The conclusion is good, but it would be nice to include a discussion on the differences between BLOB1.0 and BLOB2.0. How do the different drivers lead to different impacts?
- Detailed information on the dataset and a few links are missing.

DETAILED COMMENTS:

Introduction:

- L. 38: Do three standard deviations above normal refer to monthly values? Add an absolute temperature anomaly to give more perspective.
- L. 54: Explain in a few words why a decrease in wind-forced Ekman current leads to warm anomalies.
- L. 60: Specify which "mean flow" and clarify why it would have opposite effects in winter than summer.
- L.62: Do you mean an anomalous summertime ridging with the same characteristics than wintertime? The whole paragraph is confusing.
- L. 75: Again, I would not call them gridded observational products.

Methods:

- L. 294: NCEP GODAS does not provide "observed ocean data" but an ocean analysis resulting for a model assimilating observations. Information on the spatial resolution and the output frequency is missing.

- L. 301: Why did you choose a different product for wind speed?
- L. 303: OISST is an observational product. However, it has been shown to exhibit large bias in recent years due to changes in satellite products (Fiedler et al., 2019). Can you comment? Did you use the daily dataset?
- L. 305: the link is useless; I could not find the data.
- L. 306: Please specify how the anomalies are calculated: are these monthly anomalies? Is it consistent over the whole manuscript?
- L. 322: did you use GODAS data for both SSTA and SLPA?
- L. 328: Specify the "Pacific Meridional Mode" for the PMM.
- L. 332: "in THE Pacific Ocean"
- L. 338: Using OISST in the experiments might also be a problem (Fiedler et al., 2019). Comments?
- L. 350: Should you have detrended the monthly means of OISST for the control?

Results:

- L. 96: What do you mean by "area-weighted average"? How do you weight it?
- L. 108: Add a few words here to explain what a negative North Pacific Subtropical High Index means.
- L. 109: Specify that the y-axis in Fig. 2b is upside-down.
- L. 126: This is not in Supplementary Material.
- L. 132: Reference the equation.
- L. 144: When not shown please specify it.
- L. 154: I could not find a link to the data of NPO index. I think this should be in the Methods.
- L. 158: Typo "shift has been".
- L. 175: I don't see a negative SLPA over the Philippines in the observations Fig. 2.a.
- L. 178: Differences between observations and model outputs are not necessarily indicating internal variability, but more likely an imperfect model...
- L. 182: "Remarkably consistent" between Fig 2a and Fig 5b is an overstatement. I feel like the patterns in the North Pacific SST forced (Fig 5c) or the best member Fig. 5e are more consistent with observations.
- L. 193: Do you really feel like Fig. S2a and Fig. S2d show consistency for the precipitations?
- L. 193-195: Too fast.
- L. 223: How much reduction in low cloud? Add a symbol in Fig. S4.
- L. 226: Not shown. Figure S4 refers to the beginning of the sentence.

Summary and Discussion:

- L. 277: add a reference to Fig. 1b.

Figures:

- Fig. 1: add a mark for BLOB1.0. Add SST observations (not only the GODAS ocean reanalysis).
- Fig. 1: a) the coastline is hard to see, modify the colour. b) Note in the caption that the y-axis for the North Pacific High Intensity index is reversed. Specify which box was used for the latter.
- Fig 4: Specify "black boxes represent areas of interest of the time-series in b) and c)."

Supplementary Material:

- Fig. S1 is not necessary.
- Note: the section on "Low-cloud radiative effect vs. wind speed-induced latent heat flux in Blob 2.0 region" is interesting.

Reference:

Fiedler E.K., McLaren, A., Banzon, V., Brasmet, B., Ishizaki, S., Kennedy, J., Rayner, N., Roberts-

Jones, J., Corlett, G., Merchant, C.J., Donlon, C. (2019) Intercomparison of long-term sea surface temperature analyses using the GHRSSST Multi-Product Ensemble (GMPE) system, *Rem Sens Env*, 222, 18-33. <https://doi.org/10.1016/j.rse.2018.12.015>

Reviewer #3 (Remarks to the Author):

Review of "Physical drivers of the summer 2019 North Pacific marine heatwave-the Blob 2.0" by Amaya et al.

Comments

Marine heat waves (MHWs) are receiving considerable attention, including the current event in the North Pacific Ocean that is following on the heels, from a climate perspective, the severe MHW of 2014-16. Many readers of *Nature Communications* should be interested in the mechanisms responsible for the development of the latest North Pacific MHW. More specifically, I feel that the novel portion of this study is its finding that the sea surface temperature (SST) anomalies in the eastern subtropical North Pacific served to weaken the North Pacific High. The atmospheric model simulations based on the North Pacific SST anomalies, in other words the forced simulations lacking tropical SST anomalies, are certainly suggestive. I believe the authors pitched the importance of this mechanism about right in that it should probably be considered more a contributing rather than dominant factor in the development of the atmospheric circulation anomalies that resulted in enhanced heating of the upper ocean. The writing and graphics are clear, and the paper overall is admirably succinct. I do have some comments, in no particular order of importance, that I encourage the authors to consider in revision.

1. Latitudinal Differences – The text would benefit from a more complete recognition that the mix of mechanisms important to the development of positive SST anomalies in the sub-tropics is different from that at higher latitudes. For example, I expect that the easterly wind anomalies north of roughly 45 N, and associated poleward Ekman transport anomalies were important in the northern portion of the domain.

2. Distinction between anomaly growth and maintenance – In an interesting aspect of these events is whether the cause(s) of their development mimic those for their maintenance. It was published just recently, but Schmeisser et al. (JGR, 2019) focuses on the maintenance of the 2014-16 MHW, and the results from that paper have relevance to the present study. In particular, once the upper ocean becomes anomalously warm, the surface flux anomalies will tend to be cooling, all other factors being more typical. In the case of the 2014-16, this cooling effect was countered by extra radiative heating due to reduced cloud cover. It might be tricky for the shorter (at least so far) event of interest here, but it would be worthwhile to determine whether something can be said about what was happening at the start of the event versus after it developed.

3. Hydrostatic effects on the weakening of the North Pacific High – It would be straightforward to determine approximately how much of the observed low SLP anomaly can be attributed to a warmer atmospheric boundary layer (the Lindzen and Nigam, 1987 framework). My very rough estimate is something like 1 hPa, considerably less than what actually occurred, but the authors should look more into this. One of the tenets of the Lindzen and Nigam model is that the perturbation vanishes aloft at 700 hPa. Was this the case? Moreover, I expect that model does not work well at all north of 45 N or so. At those latitudes, seasonal mean perturbations tend to increase with height. In other words, the flow aloft is in control of the variability, with some expression in terms of the SLP.

4. Entrainment/residual term in the mixed layer heat budget – I was struck by the long-term trend in this term shown in Figure 3. I realize it represents a residual based on GODAS fields, and is prone to uncertainty and error. That the cooling rate due to this term in recent years is roughly double that early in the period deserves some scrutiny. Long-term trends in the winds and hence mixing appear minimal. Can the increase in cooling simply be attributed to the entrainment being distributed over a thinner layer with time, or is the trend more related to inadequacies/errors in the GODAS analysis (e.g., its characterization of the upper ocean mixed layer). At the very least the authors should better recognize the caveats associated with heat budgets estimated using GODAS and other ocean reanalysis products.

**Reviewer #1 (Remarks to the Author): Physical drivers of the summer 2019 North Pacific**
**marine heatwave—the Blob 2.0 by Amaya et al.**

This paper looks at the characteristics and drivers of a recent marine heatwave in the north
Pacific. The authors use a heat budget to identify the local processes that lead to the event and
bespoke model experiments to identify the influence of remote drivers. The manuscript is well
written and relates to a scientifically interesting problem. Before publication however, I feel that
a few issues need to be clarified.

A couple of general issues:
- I think there are two issues with the heat budget that at the very least require some discussion.
If I understand correctly the budget is conducted over 3-month timesteps. Heat budgets
conducted offline and particularly using long term averaged fields are likely to contain
substantial errors as the alias high-frequency processes i.e. it is very difficult to close the budget.
As such it's hard to say what component of the residual related to vertical entrainment. Related
to this, I believe the GODAS reanalysis includes data assimilation. This implies non-physical
heat sources/sinks to match the model with observations. If these terms are large it would make
any interpretation of a heat budget problematic.

- Thank you for your comment. We agree with your assessment of the potential errors and
44 uncertainties that may exist in our heat budget analysis. We also note that our
methodology is not without precedent when considering large-scale climate variability
(Bond et al. 2015; Cook et al. 2018). In response to your and other reviewer's concerns,
we have added a new subsection to the Methods called "Potential data errors and
uncertainties" to disclose the uncertainties that may arise from our methods and choice of
data sets.

*Bond, N. A., Cronin, M. F., Freeland, H. & Mantua, N. Causes and impacts of the 2014*
*warm anomaly in the NE Pacific. *Geophys. Res. Lett.* **42**, 3414–3420 (2015).*

*Cook, K. H., Vizy, E. K. & Sun, X. Multidecadal-scale adjustment of the ocean mixed*
*layer heat budget in the tropics: examining ocean reanalyses. *Clim. Dyn.* **50**, 1513–1532*
*(2018).*

- Its important to include an indication of what anomalies are statistically significant in the
composite maps for the model experiments.

- Thank you for catching this oversight. Figures 5a-c and 6b have been updated to plot only
62 95% significant anomalies. The respective captions have also been updated to make this
clearer.

Specific comments:

65: Im surprised you don't cite Di Lorenzo and Mantua 2016 here

- Thank you for catching this oversight. The reference has been added to this sentence.

98: are not only more significant ...
What do you mean by significant. To me 'significant' would relate to its impacts, however I
presume you are just saying that SSTA was larger.

• Thank you for the comment. You are correct; we meant to imply the SSTAs are larger.
We have edited the text to make this point clearer.

Fig. 2: it would be very helpful to include a contour showing the core of the MHW (based on Fig
1)

• Thank you for the excellent suggestion. We have edited Fig. 2 to include select isotherms
from Fig. 1a in order to outline the core of the MHW.

108: Subtropical High Intensity index

The use of the Subtropical High Intensity index seems somewhat circular to me given that the
index maximizes the combined SLPA and SSTA variance. It would be much cleaner to use an
SLPA or wind speed index.

- • Thank you for the comment. We agree that it would be better to compare to a subtropical
high intensity index that is independent of SST. We have edited Fig. 2 to include a North
Pacific High Intensity index based on Schmidt et al. (2019). The Methods and the
relevant text in the Results sections have all been updated to reflect the new index.

120: the residual would also include a possibly substantial error related to performing the
calculations offline using averaged data and from the fluxes associated with the data
assimilation.

- • Thank you for the comment. As mentioned above, we have added text to the Methods
section to disclose the uncertainties that may arise from performing the heat budget
calculations offline.

147 BOREAL winter

- • Thank you for the suggestion. We have edited the text to make this distinction.

Fig 5: Are these also JJA averages, as in Fig 2?

- • Yes, Fig. 5 shows the JJA 2019 average SLPA field in the AGCM experiments. We have
edited the Fig. 5 caption to make this clearer.

173: that resembles the observed weakening ...

The simulated response is far weaker than observed. Indicating that SST forcing can only
account for a small fraction of observed circulation response

- • Thank you for the comment. You are correct; the model response is much weaker than
the observations, which indicates a primary role for internal variability in generating

these anomalies. We feel this point was adequately addressed in the manuscript text in
the two paragraphs starting with “While the AGCM ensemble mean results...” and again
in the Summary and Discussion.

174: also seen in observations

In contrast to the above the simulated low over the Philippines is far weaker than in
observations. Size matters – so these discrepancies should be noted

• Thank you for the comment. Comparing Fig. 2 and Fig. 5, our AGCM experiments tend
to simulate a much *larger* low over the Philippines than seen in the observations as
opposed to a “far weaker” response that you suggest. Nevertheless, we agree with the
spirit of your comment and have added text to the manuscript to highlight these
discrepancies in the text.

193-195. This may be the case, but your experiments cant rule out that the SLP response is
purely related to local SST forcing, rather than to a teleconnection associated with a shift in the
ITCZ

• Thank you for the comment. We have added text to the end of this paragraph to note this
concern and to point to the “Local SST forcing and air-sea feedbacks” section, which
addresses the role of local SSTs in more detail.

279: Marine heatwaves are projected to become more frequent in a warmer world, which could
account for these observed changes.

This sentence doesn’t seem to make logical sense. How can future projections account for past
changes?

• Thank you for the comment. We agree that this sentence is confusing. We have edited it
to read, “Marine heatwaves frequency has been shown to increase due to past and future
global warming, which could account for these observed changes.”

**Reviewer #2 (Remarks to the Author):**

The manuscript focuses on a recent marine heatwave event in the North Pacific, which occurred
in summer 2019, the “BLOB2.0”, as opposed to the winter-time event “BLOB1.0” in 2013-2014.
The authors conclude that the persistent SST anomalies were driven by a weakening of the North
Pacific High pressure system with a significant influence of local SST feedback (through
reduced low-cloud coverage). Remote SST variability from both the tropical and subtropical
North Pacific is also shown to have contributed to the anomalous weakening of the pressure
system.

The work is original and significant, with clear motivation and discussion. The data and
methodology seem appropriate, although the agreement between Sea Level Pressure anomaly
observations and modelling experiments is overstated and not always convincing. This is the
main issue of the manuscript, and probably more a question of wording than un-conclusive
results. I recommend publication after major revision and my comments are included below.

**GENERAL:**

• The distinction between observations and model outputs is important. NCEP GODAS is not an
observational product. The first figure should include satellite SST anomalies, and not only from
OISSTv2 product which has been shown to be significantly biased in recent years (Fiedler et al.,
2019).

• Thank you for the comments and suggestions. We have edited the manuscript text
throughout to restrict our classification of “observations” to only include satellite-derived
data.

• Thank you also for bringing Fiedler et al. (2019) to our attention. The OISSTv2 data set
was chosen because it is based on one of the most comprehensively evaluated long
timeseries of SSTs, the pathfinder v5, that have shown themselves over the last decade to
have corrected for many of the offsets present when creating timeseries from drifting
orbit polar satellites. Additionally, OISSTv2 uses longer time and space scales than some
of the other data sets, which is ideal for our large-scale analysis.

• We agree that it is important to acknowledge the known biases and errors present in our
chosen datasets. Based on Figs. 4 and 5 of Fiedler et al. (2019), it would appear that the
OISSTv2 data used in the study may have biases of about 0.1-0.5°C relative to *in situ*
measurements. While these biases are admittedly significant on smaller spatial scales and
shorter time scales, they are small relative to the widespread Blob 2.0 anomalies that we
use to force our AGCM experiments, which are in excess of 2.5°C in some regions.
Additionally, our primary focus is on large-scale ocean-atmosphere variability on
seasonal-to-interannual timescales where such errors are less likely to be more important
than potential errors introduced by imperfect physics in our AGCM. Therefore, we do not
expect our AGCM results to be unduly impacted by the OISSTv2 biases.

- Below is a figure showing different SSTA timeseries averaged in the Blob 2.0 region. Data are taken from monthly mean OISSTv2 (blue), monthly mean GODAS ocean reanalysis (red), and daily data averaged to monthly means from the Canadian Meteorological Center (CMC; green). Note that data from CMC were ranked 1st in every category among data compared by Fielder et al. (2019) (see Table 5). Therefore, it offers a good standard with which to compare OISSTv2 and GODAS reanalysis against.
- While there are some differences between each data set, these differences are extremely small compared to the large overall seasonal-to-interannual variability. As a result, we have confidence in using OISSTv2 and GODAS data in this large-scale analysis.

Fig. R1 SSTs averaged in the Blob 2.0 region shown in Fig. 1a of the manuscript. Data are taken from monthly mean OISSTv2 (Jan 1982-Aug 2019; blue), monthly mean GODAS ocean reanalysis (Jan 1982-Aug 2019; red), and daily data averaged to monthly means from CMC (Jan 1992-Dec 2016; green). A common climatology of 2001-2016 was used for each data set to remove the seasonal cycle.

- To further address your comments as well as those of the other reviewers, we have added text in the Methods section to outline the uncertainties in our analysis that may arise as a result of: 1. Biases in OISSTv2, 2. Data assimilation errors in GODAS, and 3. Model uncertainties from our chosen AGCM.
- Adding all the components of the heat flux budget in supplementary materials would improve the manuscript.
- Thank you for the suggestion. We have added a figure to the Supplementary Materials to show the individual heat flux terms.
- As stated above, the comparison of observed and modelled SLPAs is not convincing: a few statements are hard to follow when looking at the figures (see below).

- Thank you for the comment. We have toned down the stronger claims in our interpretation of the AGCM results and we have adjusted the wording throughout the manuscript to make our figure descriptions clearer.

- The conclusion is good, but it would be nice to include a discussion on the differences between BLOB1.0 and BLOB2.0. How do the different drivers lead to different impacts?

- Thank you for the suggestion. We have added text to the paragraph in the Summary and Discussion section starting with “These results highlight the seasonal differences...” to focus more on how the different mechanisms discussed in this paper may lead to different impacts.

- Detailed information on the dataset and a few links are missing.

- Thank you for the comment. We have added the missing data set links to the Methods section.

DETAILED COMMENTS:

Introduction:

- L. 38: Do three standard deviations above normal refer to monthly values? Add an absolute temperature anomaly to give more perspective.

- Thank you for the suggestion. We have added an absolute temperature anomaly to the main text to give more perspective.

- L. 54: Explain in a few words why a decrease in wind-forced Ekman current leads to warm anomalies.

- Thank you for the suggestion. We have added a few words to Line 54 to make this topic clearer.

- L. 60: Specify which “mean flow” and clarify why it would have opposite effects in winter than summer.

- Thank you for comment. We have added clarifying text to this and the surrounding sentences to make the winter vs. summer distinction clearer.

- L.62: Do you mean an anomalous summertime ridging with the same characteristics than wintertime? The whole paragraph is confusing.

- Thank you for comment. We agree that the wording in this and the preceding paragraph were very confusing. In response to this and your previous comment, we have edited the manuscript text to make our point clearer. We feel the new text does a better job of outlining the different atmospheric and oceanic responses to an atmospheric ridge in winter vs summer.

• L. 75: Again, I would not call them gridded observational products.

- Thank you for comment. In response to this and your general comment above, we have edited the text to read, “In this study, we use gridded reanalysis products combined with satellite-derived observations to investigate the physical drivers...”

Methods:

• L. 294: NCEP GODAS does not provide “observed ocean data” but an ocean analysis resulting from a model assimilating observations. Information on the spatial resolution and the output frequency is missing.

- Thank you for comment. We have edited the Methods and main text throughout to more correctly refer to GODAS as an ocean reanalysis product. Additionally, we now list the GODAS spatial resolution and output frequency in the Methods.

• L. 301: Why did you choose a different product for wind speed?

- Thank you for the question. We used NCEP Reanalysis 1 for wind speed because NCEP Reanalysis 2 does not offer wind speed as an output at any time frequency.

• L. 303: OISST is an observational product. However, it has been shown to exhibit large bias in recent years due to changes in satellite products (Fiedler et al., 2019). Can you comment? Did you use the daily dataset?

- Thank you for the question. In this study, we used the monthly mean OISSTv2 output available at <https://www.esrl.noaa.gov/psd/data/gridded/data.noaa.oisst.v2.html>. The Methods have been updated with the correct link. We have added text to the Methods section to note the biases uncovered by Fiedler et al. (2019). However, as we stated above, we do not expect these errors to significantly impact our results.

• L. 305: the link is useless; I could not find the data.

- Thank you for catching this oversight. We have updated the Methods to include the proper link to the OISSTv2 data.

• L. 306: Please specify how the anomalies are calculated: are these monthly anomalies? Is it consistent over the whole manuscript?

- Thank you for the questions. We have added the following line in the Methods to make this clearer, “All anomalies reported in this manuscript are based on monthly mean data after removing long-term monthly averages.” Each subsection of the Methods reports what time period the long-term monthly averages were calculated over for each dataset.

• L. 322: did you use GODAS data for both SSTA and SLPA?

• In response to other reviewer comments, we have opted to use a new North Pacific High
Intensity index that is independent of SST. In this case, we use only NCEP Reanalysis 2
SLP data to calculate our Intensity index. The Methods have been updated accordingly.

• L. 328: Specify the “Pacific Meridional Mode” for the PMM.

• Thank you for the suggestion. We have incorporated this change in the text.

• L. 332: “in THE Pacific Ocean”

• Thank you for the suggestion. We have incorporated this change in the text.

• L. 338: Using OISST in the experiments might also be a problem (Fiedler et al., 2019).
Comments?

• Thank you for the comment. As we discuss above, it is unlikely that the OISSTv2 biases,
which are on order 0.1-0.5K, will have a large impact on our AGCM results considering
we are forcing our model with large SST anomalies in excess of 2.5°C in some regions
and considering our primary focus is on large-scale atmospheric responses.

• L. 350: Should you have detrended the monthly means of OISST for the control?

• Thank you for the question. We chose not to detrend the OISST data for the control
because we wanted to be as consistent as possible with the 2019 experiments, which were
also not detrended. However, we expect this to have a negligible impact on the results.
Figure R2 shows the non-detrended minus detrended OISST monthly climatology. The
largest differences between the two possible climatology choices are no larger than 0.05K
for any given month.

Fig. R2 Difference in the OISSTv2 monthly climatology for non-detrended vs detrended data. Shading represents the non-detrended minus detrended difference for each long-term monthly average.

Results:

• L. 96: What do you mean by “area-weighted average”? How do you weight it?

- Thank you for the question. “Area-weighted average” is a common way to refer to a latitude weighting. This weighting is done so that smaller grid points nearer to the poles are not overrepresented when taking horizontal averages.

• L. 108: Add a few words here to explain what a negative North Pacific Subtropical High Index means.

- Thank you for the suggestion. As mentioned above, we are now utilizing a different a North Pacific High Intensity index. We have incorporated your suggestion for this new index; however, we have opted to add this text in Fig. 2 caption and the Methods section.

• L. 109: Specify that the y-axis in Fig. 2b is upside-down.

- Thank you for the suggestion. We have added text to the Fig. 2 caption to note the flipped y-axis. This was done to highlight the correlation between the North Pacific Subtropical High Index and the North Pacific SSTA Index.

• L. 126: This is not in Supplementary Material.

- Thank you for the comment. We have added a figure to the supplement to show the net heat flux broken down into its components.
- L. 132: Reference the equation.
- Thank you for the suggestion. We have added the following citation for this line:
Simpson, J. H. & Bowers, D. Models of stratification and frontal movement in shelf seas. *Deep Sea Res. Part A, Oceanogr. Res. Pap.* **28**, 727–738 (1981).
- L. 144: When not shown please specify it.
- Thank you for the suggestion. We have now made this indication where appropriate.
- L. 154: I could not find a link to the data of NPO index. I think this should be in the Methods.
- Thank you for the suggestion. We have added our formulation of this index to the Methods section.
- L. 158: Typo “shift has been”.
- Thank you for catching this typo. It has been fixed.
- L. 175: I don’t see a negative SLPA over the Philippines in the observations Fig. 2.a.
- Thank you for the comment. We have changed the text to specify that the SLPAs in reference are “north of the Philippines near Taiwan”, which is consistent with the model simulations and NCEP Reanalysis 2. Additionally, we have added text to make note of the magnitude discrepancy between the observed and simulated SLPAs.
- L. 178: Differences between observations and model outputs are not necessarily indicating internal variability, but more likely an imperfect model.
- Thank you for the comment. We’ve added text to the Methods section to note how model uncertainties may impact our results and interpretations.
- L. 182: “Remarkably consistent” between Fig 2a and Fig 5b is an overstatement. I feel like the patterns in the North Pacific SST forced (Fig 5c) or the best member Fig. 5e are more consistent with observations.
- Thank you for the comment. We have removed “remarkably” from the text at this line.
- L. 193: Do you really feel like Fig. S2a and Fig. S2d show consistency for the precipitations?

• Thank you for the comment. We feel that the results in Fig. S2d are qualitatively
consistent with Fig. S2a in that both panels depict anomalies that would indicate a
general northward shift in the ITCZ. This is consistent with the Summer Deep
Convection response outlined in the text and the cited literature.

• L. 193-195: Too fast.

• Thank you for the comment. We feel that a more detailed description of the SDC
response is beyond the scope of this study. For brevity, we direct readers to the cited
literature on this topic.

• L. 223: How much reduction in low cloud? Add a symbol in Fig. S4.

• Thank you for the suggestion. A symbol has been added to now Fig. S5 to indicate the
JJA 2019 low-cloud fraction anomalies averaged in the Blob 2.0 region discussed in the
Supplemental Text.

• L. 226: Not shown. Figure S4 refers to the beginning of the sentence.

• Thank you for the comment. We have added the wind-induced latent heat flux value
reported in the Supplemental Material in the main text to make this clearer.

Summary and Discussion:

• L. 277: add a reference to Fig. 1b.

• Thank you for the suggestion. We have added the suggested reference to this line.

Figures:

• Fig. 1: add a mark for BLOB1.0. Add SST observations (not only the GODAS ocean
reanalysis).

• Thank you for the suggestion. We have added markers to represent the two peaks of Blob
1.0 in Jan-Mar 2014 and May-July 2015. However, based on the close agreement
between OISSTv2, GODAS, and CMC over their overlapping time periods and OISSTv2
and GODAS during summer 2019 (Fig. R1), we prefer to only plot data from GODAS.
We feel that plotting multiple datasets would make Fig. 1 unnecessarily difficult to
decipher and would add little to the overall interpretation of the results. We instead refer
readers to the added text on uncertainties and potential errors in the Methods section.

• Fig. 2: a) the coastline is hard to see, modify the colour. b) Note in the caption that the y-axis
for the North Pacific High Intensity index is reversed. Specify which box was used for the latter.

• Thank you for the suggestions. We have thickened the coastline contour in an effort to
make it more visible. We now also note the reversed y-axis in panel b in the Fig. 2
caption. We refer readers to the Methods section when discussing the formulation of the
North Pacific High Intensity index.

• Fig 4: Specify “black boxes represent areas of interest of the time-series in b) and c).”

- Thank you for the suggestion. We have added the suggested text to the Fig. 4 caption.

Supplementary Material:

• Fig. S1 is not necessary.

- Thank you for the comment. This figure was included in response to multiple requests to see it when presenting this work at workshops and conferences. We prefer to keep it in the Supplementary Material.

• Note: the section on “Low-cloud radiative effect vs. wind speed-induced latent heat flux in Blob 2.0 region” is interesting.

- Thank you!

Reference:

Fiedler E.K., McLaren, A., Banzon, V., Brasnett, B., Ishizaki, S., Kennedy, J., Rayner, N., Roberts-Jones, J., Corlett, G., Merchant, C.J., Donlon, C. (2019) Intercomparison of long-term sea surface temperature analyses using the GHRSSST Multi-Product Ensemble (GMPE) system, Rem Sens Env, 222, 18-33. <https://doi.org/10.1016/j.rse.2018.12.015>

**Reviewer #3 (Remarks to the Author):**

**Review of “Physical drivers of the summer 2019 North Pacific marine heatwave-the Blob**
**2.0” by Amaya et al.**

Comments

Marine heat waves (MHWs) are receiving considerable attention, including the current event in
the North Pacific Ocean that is following on the heels, from a climate perspective, the severe
MHW of 2014-16. Many readers of Nature Communications should be interested in the
mechanisms responsible for the development of the latest North Pacific MHW. More
specifically, I feel that the novel portion of this study is its finding that the sea surface
temperature (SST) anomalies in the eastern subtropical North Pacific served to weaken the North
Pacific High. The atmospheric model simulations based on the North Pacific SST anomalies, in
other words the forced simulations lacking tropical SST anomalies, are certainly suggestive. I
believe the authors pitched the importance of this mechanism about right in that it should
probably be considered more a contributing rather than dominant factor in the development of
the atmospheric circulation anomalies that resulted in enhanced heating of the
upper ocean. The writing and graphics are clear, and the paper overall is admirably succinct. I do
have some comments, in no particular order of importance, that I encourage the authors to
consider in revision.

1. Latitudinal Differences – The text would benefit from a more complete recognition that the
mix of mechanisms important to the development of positive SST anomalies in the sub-tropics is
different from that at higher latitudes. For example, I expect that the easterly wind anomalies
north of roughly 45 N, and associated poleward Ekman transport anomalies were important in
the northern portion of the domain.

• Thank you for the suggestion. We have added a paragraph to the Summary and
Discussion section to discuss latitudinally dependent mechanisms and their role in
shaping the JJA 2019 SSTA pattern discussed in this study.

2. Distinction between anomaly growth and maintenance – In an interesting aspect of these
events is whether the cause(s) of their development mimic those for their maintenance. It was
published just recently, but Schmeisser et al. (JGR, 2019) focuses on the maintenance of the
2014-16 MHW, and the results from that paper have relevance to the present study. In particular,
once the upper ocean becomes anomalously warm, the surface flux anomalies will tend to be
cooling, all other factors being more typical. In the case of the 2014-16, this cooling effect was
countered by extra radiative heating due to reduced cloud cover. It might be tricky for the shorter
(at least so far) event of interest here, but it would be worthwhile to determine whether
something can be said about what was happening at the start of the event versus after it
developed.

• Thank you for the suggestion. Below is a figure that shows the surface latent heat flux
(blue) and the net surface downward shortwave radiation (orange) averaged in the Blob
2.0 region for January-August 2019. Also plotted are the Blob 2.0 SSTAs for the same

time period. Note that there are currently no reliable surface shortwave data for our time
 period of interest, so we've again estimated shortwave radiation using MODIS low-cloud
 fraction anomalies (see Fig. S5 for an example and the Supplemental Methods for more
 details). For each month, we multiply the respective regression coefficient by the MODIS
 low-cloud fraction anomaly to get an estimate of the change in shortwave radiation due to
 the change in low-clouds.

**Fig. R3** Data averaged in the Blob 2.0 region for the period January-August 2019. Left axis:
 NCEP2 latent heat flux (blue; positive= ocean warming) and satellite estimated net downward
 surface shortwave radiation (orange; see Supplemental Materials). Right axis: GODAS 5m
 temperature anomalies. Note that all data was subject to a 3-month running mean before plotting.

- • Figure R3 shows the relative importance of different heat flux terms during the growth
 and peak of Blob 2.0. During the growth of the event, latent heat flux associated with the
 weakened atmospheric circulation described in Fig. 2 is much stronger than the
 shortwave term. However, as SSTAs continue to warm, increased shortwave due to
 changes in low-clouds appears to be more important. Based on these results, it's possible
 that changes in surface winds primarily acted as a “trigger” for Blob 2.0, while low-cloud
 SST feedback played a larger role in continuing to grow and maintain the SSTAs.
- • While Fig. R3 is certainly suggestive, we feel that the lack of reliable shortwave radiation
 data during this period makes it difficult to assess this topic much further. We prefer to
 not include Fig. R3 in the manuscript, however, text has been added to the “Local SST
 forcing and air-sea feedbacks” section to briefly discuss these findings and to call upon
 future research on this topic once more data becomes available. We have also included a
 reference to Schmeisser et al. (2019).

3. Hydrostatic effects on the weakening of the North Pacific High – It would be straightforward
 to determine approximately how much of the observed low SLP anomaly can be attributed to a
 warmer atmospheric boundary layer (the Lindzen and Nigam, 1987 framework). My very rough
 estimate is something like 1 hPa, considerably less than what actually occurred, but the authors
 should look more into this. One of the tenets of the Lindzen and Nigam model is that the
 perturbation vanishes aloft at 700 hPa. Was this the case? Moreover, I expect that model does not

work well at all north of 45 N or so. At those latitudes, seasonal mean perturbations tend to
 increase with height. In other words, the flow aloft is in control of the variability, with some
 expression in terms of the SLP.

- • Thank you for the comments and suggestions. Below is a figure that shows JJA 2019
 geopotential height anomalies at different levels in NCEP Reanalysis 2 (NCEP2) and the
 ensemble mean of our North Pacific only AGCM experiments. It seems the observed and
 AGCM height perturbations vanish above 700 hPa south of about 40°N, which is
 consistent with the Lindzen and Nigam model.

**Fig. R4** JJA 2019 averaged geopotential height anomalies (m) in: (top row) NCEP2 and (bottom
 row) the ensemble mean of our North Pacific only AGCM experiment. Columns represent
 anomalies at 1000 hPa, 700 hPa, and 500 hPa from left to right, respectively.

- • Using the Lindzen and Nigam (1987) model, we can estimate the hydrostatic effect on
 SLP (p') as:

$$584 \quad p' = g\rho_0 n H_0 \left(\frac{\gamma}{2} - 1 \right) SST' \quad (\text{Eq. R1})$$

And,
$$n = \frac{1}{T_0}$$

- • Where g is gravitational acceleration, the reference density $\rho_0=1.225 \text{ kg m}^{-3}$, the
 reference temperature $T_0=288 \text{ K}$, the reference height $H_0=3000\text{m}$, the fraction of
 perturbation remaining at H_0 is $\gamma=0.3$, and SST' are SSTAs. Figure R5 shows p' for JJA
 2019 as estimated with OISSTv2 data. We restrict our analysis to equatorward of 40°N
 where we may expect the model to work well.

Fig. R5 JJA 2019 values of p' from Eq. R1. Estimated using data from OISSTv2. Gray contours show the NCEP2 SLP map from Fig. 2a of the main manuscript. SLP contours start at 0.5 hPa with a 0.5 hPa interval.

- Based on Fig. R5, subtropical SSTAs may have contributed to ~ 1 hPa to the overall negative SLPAs in this region. We have added text to the end of the newly named “Local SST forcing and air-sea feedbacks” section to include this result. However, for brevity we have opted to not include these figures in the manuscript.

4. Entrainment/residual term in the mixed layer heat budget – I was struck by the long-term trend in this term shown in Figure 3. I realize it represents a residual based on GODAS fields, and is prone to uncertainty and error. That the cooling rate due to this term in recent years is roughly double that early in the period deserves some scrutiny. Long-term trends in the winds and hence mixing appear minimal. Can the increase in cooling simply be attributed to the entrainment being distributed over a thinner layer with time, or is the trend more related to inadequacies/errors in the GODAS analysis (e.g., its characterization of the upper ocean mixed layer). At the very least the authors should better recognize the caveats associated with heat budgets estimated using GODAS and other ocean reanalysis products.

- Thank you for the comments and suggestions. Given that our heat budget is based on long term (3-month) averages and was computed “offline”, we feel that it may be inappropriate to overinterpret the physical meaning behind the residual and its trend. Additionally, we feel that doing so is beyond the scope of this study and would require a more targeted research focus.

- In response to your and other reviewer’s comments, we have added text to the Methods section to acknowledge the potential uncertainties and errors that may arise in our heat budget due to our chosen methods and data set. However, we defer to future studies to investigate this topic further.

REVIEWERS' COMMENTS:

Reviewer #1 (Remarks to the Author):

The authors have gone to considerable efforts to respond in detail to my comments and I find the manuscript much improved and feel that it will make a useful contribution to the field of marine heatwaves. I do still have a few relatively minor issues that should be considered.

120: May to August

It would be helpful to state the motivation for using May to August?

Fig. 3 – it is somewhat misleading to join the dots in these figures. You are not showing continuous temperature tendency, each year is independent (i.e. the starting point in May could be a very different value each year). A bar blot would be more appropriate.

147 with THE exception

Fig S2b shows a cooling effect associated with latent heat not a warming – is there a mistake with the sign on this plot?

I find the ML budget description still lacks some detail. Are you calculating the budget for each month between May and August (i.e. using monthly varying heat fluxes, velocities and mixed layer depth) and then showing the average of these for MJJA in Figs 3 and S2?

It would be interesting to know what the mixed layer (and wind speeds) were doing prior to the period of maximum warming. If the region had a shallow mixed layer prior to the build up of the MHW, this would precondition the ocean for enhanced warming during summer when the net radiative flux becomes positive.

165: CP ENSO events energize the NPO 24,25

The references cited here appear to be talking about much longer (decadal) timescales than those here.

166: This is even more evident ...

This statement suggests that you are going to provide more evidence the CP ENSO links to NPO

177: We test this hypothesis ...

Prior to delving into models wouldn't it make sense to show that these linkages exist in the observations? As noted above literature cited doesn't really suggest that the CP/NPO relationship exists on timescales relevant for this work and haven't tied PMM changes to the atmospheric patterns observed. Would it be helpful to show a correlation between CP SSTA and SLP and PMM vs SLP during summer to demonstrate these relationships?

185: I think it is necessary at this point to acknowledge that the magnitude of the forced signal is considerably weaker than observed. You could, for example, quantify this by providing the NP high index used previously. It looks like the SST forcing could explain about one third of the pressure response.

199: CP El Niño conditions

Im not sure that Id call this a CP El Nino. Its warm SSTA in the CP region. CP El Nino would not be occurring at this time of year.

209: Fig. 5f.

Do you mean 5c?

220-228

Why use correlation in this black box region? There's no reason to expect that if the synoptic conditions are similar to observed in the NE Pacific that they would also be similar over the SW Pacific. The MHW is associated to a much more local weakening of the NE Pacific high. I would think that it would be more useful to see if any of the ensemble members capture a weakening of the NE Pacific high that is of a similar magnitude to the observations?

Indeed if you increase the number of ensemble members you are going to find ensemble members that by chance have a higher correlation to observations. But this would be true even if you didn't use the same SST forcing. So, I'm not really sure what useful information this provides. As I suggest above, perhaps a more useful question is can the forcing + internal variability change the NE Pacific high to the same extent as observed? Or even better, how often can the forcing + internal variability change the NE Pacific high to the same extent as observed?

222: closely captures ...

I think this description is not really accurate particularly for the total simulation

250-252: Whereas, the peak latent heat flux anomalies precede the largest shortwave anomalies by several months, and may be thought of as a "triggering" mechanism for this event brought about by the weakened North Pacific High...

This begs the question: wouldn't it have been more useful to present the heat budget analysis by month, rather than showing the difference between May and August? This would then have shown the relative timing of the different terms.

296: Without sustained forcing, it is unlikely Blob 2.0 will produce as significant damage to marine ecosystems as its predecessor

While Blob2 would likely be shorter because of factors you describe, the SST anomalies are occurring in summer. A hot summer could potentially have more severe impacts (due to thermal tolerances being exceeded) than a hot winter.

308: which could account for these changes.

I think the causality is mixed up here. The increase in background SST means more MHWs, not the other way around.

413: calculating the heat budget "offline" across long-term average fields may cloud interpretation of the results, particularly the mechanisms that drive variability in the residual.

I think it does more than cloud the interpretation. Use of long-term means introduce errors into the calculation. For example, even if the surface heat flux and mixed layer depth were to change linearly over the course of the averaging period the time mean of daily SHF/MLD would be different from the time mean daily SHF divided by the time mean MLD.

Unfortunately, without an independent estimate of entrainment, I don't think its possible to quantify the uncertainty in the heat budget.

Regards

Alex Sen Gupta

Reviewer #2 (Remarks to the Author):

The edited manuscript has improved substantially from the original submitted version. The authors addressed or explained each comment sufficiently. In particular the new supplementary figures and the additional section on uncertainties are great additions.

I therefore recommend the submission for Nature Communication.

I have a couple last comments. First, adding a figure on the EKE during the BLOB2.0 could help convincing the reader about the limited influence of lateral advection, which is surprisingly low in the heat budget. Second, I would like to see more explanation on the various flux calculations shown in figure S2 ("Full", "MLD" and "Flux"). I guess it is explained in the caption (which I could not find) but it deserves a few lines in the method section.

Reviewer #3 (Remarks to the Author):

Review of revised version of "Physical drivers of the summer 2019 North Pacific marine heatwave-the Blob 2.0" by Amaya et al.

The authors have adequately addressed the concerns I raised in my review of the original manuscript, and I applaud them for the timeliness of this publication.

**Reviewer #1 (Remarks to the Author):**

The authors have gone to considerable efforts to respond in detail to my comments and I find the
manuscript much improved and feel that it will make a useful contribution to the field of marine
heatwaves. I do still have a few relatively minor issues that should be considered.

120: May to August

It would be helpful to state the motivation for using May to August?

 - Thank you for the suggestion. We have added text to the manuscript to motivate this
decision.

Fig. 3 – it is somewhat misleading to join the dots in these figures. You are not showing
continuous temperature tendency, each year is independent (i.e. the starting point in May could
be a very different value each year). A bar blot would be more appropriate.

 - Thank you for the comment. As you suggest we do not intend to imply that Fig. 3
represents a continuous temperature tendency; however, we feel that joining the dots as
we've done provides a clearer depiction of the interannual variations and trends present
in our timeseries (particularly in Q_{net} , the mixed layer depth and the wind speed cubed),
which are useful and interesting to discuss. Additionally, we feel that a bar chart would
become too cluttered and unnecessarily difficult to decipher. Therefore, we prefer to
follow the lead of Bond et al. (2015), which is closely related to our work, and present the
heat budget results as a line plot with dots.

147 with THE exception

 - Thank you for catching this typo, the text has been edited.

Fig S2b shows a cooling effect associated with latent heat not a warming – is there a mistake
with the sign on this plot?

 - Thank you for the question. This is not a sign error. Fig. S2b shows the individual heat
flux terms broken down by Eq. S2 into a contribution from the time-varying heat flux
(FLUX, dotted Fig. S2) and a contribution from a time-varying mixed layer depth (MLD,
dashed Fig. S2). In JJA 2019, the contribution of the time-varying latent heat flux
(FLUX) term is positive (as we expect), while the time-varying mixed layer depth (MLD)
term is negative. The negative MLD value is the result of the extremely thin mixed layer
depth during JJA 2019. In response to this and other comments below, text has been
added to the Supplemental Materials and main manuscript to clarify the mixed layer heat
budget formulation.

I find the ML budget description still lacks some detail. Are you calculating the budget for each
65 month between May and August (i.e. using monthly varying heat fluxes, velocities and mixed
layer depth) and then showing the average of these for MJJA in Figs 3 and S2?

- Thank you for the comment and question. We have added text to the Method section to clarify our formulation of the heat budget.

It would be interesting to know what the mixed layer (and wind speeds) were doing prior to the period of maximum warming. If the region had a shallow mixed layer prior to the build up of the MHW, this would precondition the ocean for enhanced warming during summer when the net radiative flux becomes positive.

- Thank you for the suggestion. Below are the wind speeds and MLD values averaged in the Blob 2.0 region for January-August 2019 (Fig. R1). Based on this figure, it appears that the mixed layer first shoaled significantly in April, perhaps in response to negative wind speed cubed anomalies in the prior monthly. Text has been added to the “Drivers of Blob 2.0 in observational analyses” section of the manuscript to summarize these results; however, for brevity, we prefer to not include this figure.

Figure R1: Wind speed cubed (top) and MLD (bot) values averaged in the Blob 2.0 region for

January-August 2019.

165: CP ENSO events energize the NPO 24,25

The references cited here appear to be talking about much longer (decadal) timescales than those

here.

- Thank you for the comment. We have added to text in this paragraph of the manuscript make this timescale distinction with respect to these particular citations.

166: This is even more evident ...

This statement suggests that you are going to provide more evidence the CP ENSO links to NPO

- Thank you for the comment. We have edited this sentence to make our meaning clearer.

177: We test this hypothesis ...

Prior to delving into models wouldn't it make sense to show that these linkages exist in the
observations? As noted above literature cited doesn't really suggest that the CP/NPO relationship
exists on timescales relevant for this work and haven't tied PMM changes to the atmospheric
patterns observed. Would it be helpful to show a correlation between CP SSTA and SLP and
PMM vs SLP during summer to demonstrate these relationships?

• Thank you for the suggestion and questions. We have toned down the rhetoric linking CP
ENSO to NPO and have altered the text to more accurately describe the cited literature
surrounding this connection. While we agree that it would be useful to more thoroughly
explore the interannual link between CP ENSO and the NPO in observations, we believe
this analysis warrants a targeted focus that is beyond the scope of this paper. In particular,
our tropical AGCM results are suggestive of a physical connection that may motivate
future studies on this topic. Additionally, the connection between the PMM and North
Pacific atmospheric circulation in boreal summer has been heavily investigated in both
coupled models and observations (e.g., Vimont et al. 2003; Chiang and Vimont, 2004;
Amaya 2019; Amaya et al. 2019). For brevity, we refer readers to these studies.

185: I think it is necessary at this point to acknowledge that the magnitude of the forced signal is
considerably weaker than observed. You could, for example, quantify this by providing the NP
high index used previously. It looks like the SST forcing could explain about one third of the
pressure response.

• Thank you for the comment and suggestion. We address the magnitude discrepancy
between the ensemble AGCM results and the observations in detail in the paragraph
starting with "While the AGCM ensemble mean results...". We have added text to this
paragraph to report the observed and AGCM ensemble mean North Pacific High Intensity
index values for JJA 2019.

199: CP El Niño conditions

Im not sure that Id call this a CP El Nino. Its warm SSTA in the CP region. CP El Nino would
not be occurring at this time of year.

• Thank you for the comment. The Nino4 index is a commonly used metric to quantify CP
ENSO variations. Below are 3-month running mean Nino4 values from January 2018-
August 2019 in OISSTv2. If we define an El Niño event as 5 consecutive 3-month
averages above 0.5°C, it appears that CP El Niño conditions were first established in
winter 2018/2019 and then persisted into the summer 2019. Text has been added to the
manuscript to make this point.

Figure R2: The Nino4 index using OISSTv2 from January 2018 to August 2019. Data was subject to a 3-month running mean before plotting. Black line represents 0.5°C.

209: Fig. 5f.

Do you mean 5c?

- Thank you for catching this typo, the text has been edited.

220-228

Why use correlation in this black box region? There's no reason to expect that if the synoptic conditions are similar to observed in the NE Pacific that they would also be similar over the SW Pacific. The MHW is associated to a much more local weakening of the NE Pacific high. I would think that it would be more useful to see if any of the ensemble members capture a weakening of the NE Pacific high that is of a similar magnitude to the observations?

154

- Thank you for the comment. We chose this box to include the SW Pacific because the tropically-forced AGCM experiments suggest enhanced convection and the resulting low SLP in this region are related to the overall weakening of the North Pacific High (Figs. S3c and 5b). These features are also seen in the reanalysis data, although the anomalies are much weaker, which we note in the main text.
- Our results are unchanged when selecting a box that primarily focuses on the North Pacific High (Fig. R3). In this case, the same 3 single ensemble members are chosen as having the highest pattern correlations in this new box. Text has been added to the manuscript to note the insensitivity of our model comparison to the domain.

164

Figure R3: Same as Fig 2 of the main text, but highlighting a region that primarily focuses on the weakened North Pacific High.

Indeed if you increase the number of ensemble members you are going to find ensemble members that by chance have a higher correlation to observations. But this would be true even if you didn't use the same SST forcing. So, I'm not really sure what useful information this provides. As I suggest above, perhaps a more useful question is can the forcing + internal variability change the NE Pacific high to the same extent as observed? Or even better, how often can the forcing + internal variability change the NE Pacific high to the same extent as observed?

- Thank you for the comment. To address these questions, we have added text to the end of the "Remote SST forcing" section to indicate how many ensemble members produce North Pacific High Intensity index values that are equal to or more negative than the JJA 2019 observed value (-1.94 hPa; Fig. 2b).

222: closely captures ...

I think this description is not really accurate particularly for the total simulation

- Thank you for the comment. We have edited this text to say "more closely captures". We feel this statement is supported by a comparison between the single ensemble member results (Fig. 5d-f) and the ensemble mean results (Fig. 5a-c).

250-252: Whereas, the peak latent heat flux anomalies precede the largest shortwave anomalies by several months, and may be thought of as a "triggering" mechanism for this event brought about by the weakened North Pacific High...

This begs the question: wouldn't it have been more useful to present the heat budget analysis by month, rather than showing the difference between May and August? This would then have shown the relative timing of the different terms.

- Thank you for the comment. We chose to present the heat budget related to the August minus May SST difference in order to more easily compare the results to prior years and

197 thus further put Blob 2.0 into historical context. A month-to-month heat budget of only
198 2019 would have made this comparison difficult.

- • Additionally, at the time of the analysis there were no reliable observations of surface
shortwave radiation during this time period, which forced us to estimate the effect of
shortwave radiation on the surface heat budget through a linear regression technique with
MODIS satellite-derived low clouds. This technique is useful in the summertime when
shortwave radiation and low cloud fraction anomalies in this region are highly correlated,
but it begins to breakdown in the spring and winter when marine stratocumulus clouds
are less pervasive. As a result, we felt it was inappropriate to provide an in-depth
assessment of the heat fluxes prior to summer 2019 until reliable observations become
widely available. We defer to future research on this topic.

296: Without sustained forcing, it is unlikely Blob 2.0 will produce as significant damage to
marine ecosystems as its predecessor

While Blob2 would likely be shorter because of factors you describe, the SST anomalies are
occurring in summer. A hot summer could potentially have more severe impacts (due to thermal
tolerances being exceeded) than a hot winter.

- • Thank you for this excellent point. We have added text to the paragraph on impacts in the
Summary and Discussion section to highlight this distinction.

308: which could account for these changes.

I think the causality is mixed up here. The increase in background SST means more MHWs, not
the other way around.

- • Thank you for the comment. We edited this sentence to improve its clarity.

413: calculating the heat budget “offline” across long-term average fields may cloud
interpretation of the results, particularly the mechanisms that drive variability in the residual.
I think it does more than cloud the interpretation. Use of long-term means introduce errors into
the calculation. For example, even if the surface heat flux and mixed layer depth were to change
linearly over the course of the averaging period the time mean of daily SHF/MLD would be
different from the time mean daily SHF divided by the time mean MLD.

Unfortunately, without an independent estimate of entrainment, I don’t think its possible to
quantify the uncertainty in the heat budget.

- • Thank you for the comment. We agree with your assessment and have added text to the
“Potential data errors and uncertainties” section of the Methods to make this additional
point.

Regards

Alex Sen Gupta

Reviewer #2 (Remarks to the Author):

The edited manuscript has improved substantially from the original submitted version. The
authors addressed or explained each comment sufficiently. In particular the new supplementary
figures and the additional section on uncertainties are great additions.

I therefore recommend the submission for Nature Communication.

I have a couple last comments. First, adding a figure on the EKE during the BLOB2.0 could help
convincing the reader about the limited influence of lateral advection, which is surprisingly low
in the heat budget.

• Thank you for the suggestion. Figure R4 below shows the EKE for JJA 2019. Eddy
kinetic energy in the North Pacific is very weak away from the Kuroshio Current
Extension (Qiu 2001), including in our region of interest during JJA 2019 (black box).
Additionally, EKE becomes decreasingly important in the heat budget when averaged
over a large domain. We have added text and the following citation to the manuscript to
make this point; however, for brevity, we would prefer to not include Fig. R4 in the final
manuscript.

Qiu, B. The Kuroshio Extension system: Its large-scale variability and role in the
midlatitude ocean-atmosphere interaction. *J. Oceanogr.* **58**, 57–75 (2002).

Figure R4: EKE anomalies for JJA 2019. Calculated from AVISO daily mean satellite data.
Black box represents the Blob 2.0 region shown in Fig. 1a of the main text.

Second, I would like to see more explanation on the various flux calculations shown in figure S2
(“Full”, “MLD” and “Flux”). I guess it is explained in the caption (which I could not find) but it
deserves a few lines in the method section.

• Thank you for the suggestion. We have added text to the Supplemental Methods to better
explain the fluxes depicted in Figure S2.

Reviewer #3 (Remarks to the Author):

Review of revised version of “Physical drivers of the summer 2019 North Pacific marine
heatwave-the Blob 2.0” by Amaya et al.

The authors have adequately addressed the concerns I raised in my review of the original
manuscript, and I applaud them for the timeliness of this publication.

- • Thank you to you and the other reviewers for your taking the time to review our
manuscript. Each of your suggestions and comments greatly improved the clarity of our
work and led to new insights.
